# Temporal Trends in Oral Anticoagulant Prescription in Atrial Fibrillation Patients between 2004 and 2019

**DOI:** 10.3390/ijerph19095584

**Published:** 2022-05-04

**Authors:** Iwona Gorczyca-Głowacka, Bernadetta Bielecka, Paweł Wałek, Magdalena Chrapek, Agnieszka Ciba-Stemplewska, Olga Jelonek, Anna Kot, Anna Czyżyk, Maciej Pióro, Agnieszka Major, Beata Wożakowska-Kapłon

**Affiliations:** 1Collegium Medicum, Jan Kochanowski University, 25-369 Kielce, Poland; iwona.gorczyca@interia.pl (I.G.-G.); olga.jelonek@wp.pl (O.J.); czyzykania@gmail.com (A.C.); mp12111@gmail.com (M.P.); major.agn@gmail.com (A.M.); bw.kaplon@poczta.onet.pl (B.W.-K.); 21st Clinic of Cardiology and Electrotherapy, Swietokrzyskie Cardiology Centre, 25-736 Kielce, Poland; bielecka.bernadetta@gmail.com (B.B.); ania.kot.ak@gmail.com (A.K.); 3Faculty of Natural Sciences, Jan Kochanowski University, 25-369 Kielce, Poland; chrapek.magdalena@gmail.com; 4Department of Internal Medicine, Integrated Provincial Hospital, 25-736 Kielce, Poland; aciba@interia.pl

**Keywords:** atrial fibrillation, non-vitamin K antagonist oral anticoagulants, oral anticoagulants, vitamin K antagonists

## Abstract

Background: In the recent years, antithrombotic prophylaxis in patients with atrial fibrillation (AF) has changed significantly. The main aim of this study is to assess the temporal trends of antithrombotic therapy and identify factors predisposing oral anticoagulant (OAC) use in stroke prevention in AF patients. Methods: The present study is a retrospective, observational, single-center study, which includes consecutively hospitalized patients in the reference cardiology center from January 2004 to December 2019. Results: A total of 9656 patients (43.7% female, mean age 71.2 years) with AF between 2004–2019 are included. Among the total study population, in most of the patients (81.1%), OAC therapy was used, antiplatelet (APT) therapy was prescribed for 13.5% patients, heparins for 2.1% patients and 3.3% of patients did not receive any stroke prevention. OAC prescription significantly increased from 61.6% in 2004 to 97.4% in 2019. The independent predictors of OAC prescription were: the period of hospitalization, non-paroxysmal AF, age, hypertension, diabetes mellitus, previous thromboembolism, hospitalization due to electrical cardioversion, ablation or AF without any procedures. Conclusions: In hospitalized patients with AF, during sixteen years of the study period, a significant increase in OAC use and a decrease in APT use were noted. Factors other than these included in the CHA_2_DS_2_-VASc score were independent predictors of OAC use.

## 1. Introduction

Atrial fibrillation (AF) is associated with an up to five-fold increase in the thromboembolic event risk, and thus stroke prevention is important in the management of patients with AF [1,2]. Changes in antithrombotic therapy in recent years are conditioned by both the alteration of scientific association guidelines concerning the treatment of AF patients, and the introduction of non-vitamin K oral anticoagulants (NOACs). In all the guidelines, the application of oral anticoagulants (OACs) is recommended in patients with high thromboembolic risk and not recommended to patients without thromboembolic risk factors [3,4,5]. Stroke prevention in AF patients is a part of the Atrial Fibrillation Better Care (ABC) holistic pathway, where ‘A’ is defined as Anticoagulation/Avoid stroke [5]. Using the CHA_2_DS_2_-VASc score to evaluate the risk of thromboembolism has been recommended since 2010 [3]. Prior to this, it was advised that the CHADS_2_ score should be used, however there was a need to include additional factors (vascular disease, aged 65–74 years, female sex) when taking decisions concerning antithrombotic therapy use [6]. In clinical practice, the antithrombotic treatment of AF patients was significantly influenced by registering NOACs, which appeared to be at least as effective as a vitamin K antagonist (VKA), but safer [7,8,9].

The main aim of this study is to assess the temporal trends of antithrombotic therapy in the stroke prevention in AF patients. Secondly, we evaluate the antithrombotic patterns in the particular stroke risk groups, and, finally, we identify the factors predisposing patients to OAC use. 

## 2. Methods

### 2.1. Study Design and Study Population

We conducted a hospital-based retrospective study of patients aged ≥18 years and with AF admitted to the Swietokrzyskie Cardiology Center. The study was registered in ClinicalTrials.gov: NCT04419012. The Swietokrzyskie Cardiology Center is the largest referral hospital in the Swietokrzyskie province and provides specialist medical care for a population of approximately 1,230,000 people in south-east of Poland. 

The data were collected from January 2004 to December 2019. The study includes all consecutive patients with AF hospitalized during the study period for urgent and planned reasons. Patients were included if they were at least 18 years of age and had a history of AF documented by electrocardiography or in their medical history. Patients with valvular AF, death during hospitalization and with incomplete data concerning antithrombotic treatment were excluded from the study. After applying the exclusion criteria described above, a total of 9656 patients were included in this study (Figure 1). 

### 2.2. Covariaties 

Electronic medical records of all patients were used as the data source. Patients’ electronic medical records contained information including age, gender, clinical characteristics, laboratory data and antithrombotic treatment. The clinical characteristics were: stroke risk factors, bleeding risk factors, active cancer and peptic ulcer. The reasons for hospitalization were included in the analysis. The stroke risk factors were as follows (according to the CHA_2_DS_2_-VASc score): previous stroke; transient ischemic attack (TIA) or peripheral thromboembolism; heart failure; vascular disease (angiographically significant coronary artery disease, previous myocardial infarction, peripheral arterial disease or aortic plaque); hypertension; diabetes mellitus; female and age. The bleeding risk factors were as follows (according to the HAS-BLED score): hypertension; age > 65 years; stroke; previous bleeding; anemia (HGB < 12 g/dL in women, <13 g/dL in men); thrombocytopenia (PLT < 150 × 10^3^/µL); renal disease; liver disease; concomitant treatment with antiplatelet drugs (APT) or non-steroidal anti-inflammatory drugs or high alcohol intake. The definitions of diseases included in the CHA_2_DS_2_-VASc and HAS-BLED scores are presented in Appendix A.

AF was diagnosed on the basis of the definition of the European Society of Cardiology, according to which arrhythmia can be identified using an electrocardiogram showing an irregular atrial rhythm lasting longer than 30 s [5].

The glomerular filtration rate (GFR), used to assess the patients’ kidney function, was calculated using the MDRD equation.

The study was approved by the Ethics Committee of the Swietokrzyska Medical Chamber in Kielce (104/2012). The Ethics Committee waived the requirement of obtaining informed consent from the patients.

### 2.3. Stroke Risk Assessment 

The CHADS_2_ and CHA_2_DS_2_-VASc scores were calculated for all patients. The CHADS_2_ score was calculated by assigning 1 point for an age of ≥75 years, heart failure, hypertension and diabetes mellitus, and 2 points for a previous thromboembolic event; and the CHA_2_DS_2_-VASc score was calculated assigning 1 point for an age between 65 and 74 years, heart failure, hypertension, diabetes mellitus, vascular disease and female, and 2 points for a previous thromboembolic event and an age of ≥75 years [10].

During the study period, both the CHADS_2_ and CHA_2_DS_2_-VASc scores were used to assess the risk of thromboembolic complications. The CHADS_2_ score was used until 2010, and then the CHA_2_DS_2_-VASc score was recommended. In the present study, the CHA_2_DS_2_-VASc score was used to qualify for thromboembolic risk groups for two reasons: first, most of the patients in the present study were hospitalized after 2010; secondly, in ESC guidelines from 2006, the assessment of additional risk factors included in the CHA_2_DS_2_-VASc score (female, vascular disease, aged 65–74 years) was recommended. 

According to the CHA_2_DS_2_-VASc score, low-thromboembolic-risk patients were classified as having a score of 0 (1 in women), intermediate-thromboembolic-risk patients as having a score of 1 (2 in women), and high-thromboembolic-risk patients as having a score ≥2 (≥3 in women).

### 2.4. Antithrombotic Therapy among the Study Group 

The antithrombotic therapy recommended during the patients’ discharge from the hospital was evaluated. The following types of regimens were defined: OAC ± antiplatelet (APT) therapy, APT alone, heparin and no antithrombotic treatments. The OAC group included VKAs, apixaban, dabigatran, and rivaroxaban alone or with APT. Edoxaban has been registered in Europe as a drug for preventing thromboembolic complications in patients with AF, however it is not available in Poland. The APT group included acetylsalicylic acid or/and clopidogrel, ticagrelor and prasugrel. No antithrombotic treatment was defined by the absence of OAC, heparin and APT prescription. 

### 2.5. Statistical Analyses

Continuous data were described by means, standard deviations, medians and interquartile ranges. Categorical data were summarized by frequencies and percentages and group comparisons were performed using the chi-squared or Fisher’s exact test. The OAC prescription was modeled by univariable and multivariable logistic regressions, and the odds ratios (ORs) with a 95% confidence interval (95% CI) were calculated. A two-tailed *p*-value < 0.05 was considered statistically significant. All statistical analyses were performed using the R software package version 4.0.3 (Vienna, Austria).

## 3. Results 

### 3.1. Patient Characteristics 

A total of 9656 patients (43.7% female, mean age: 71.2 years) with AF between 2004–2019 were included in the study. Hypertension was the most common co-morbidity (76.5%), whereas 58.7% of patients had a concomitant diagnosis of heart failure. Among non-cardiac co-morbidities, impaired renal function was the most common (62.6%). The most commonly reported AF type was paroxysmal AF (45.5%), whereas 39.6% of patients had a permanent AF. The clinical characteristics of patients according to antithrombotic strategies are presented in Table 1.

Only 3.8% of patients were at low risk of stroke (CHA_2_DS_2_-VASc = 0 in men, 1 in women), while most patients (85.6%) were at high risk of stroke (CHA_2_DS_2_-VASc ≥ 2 in men, ≥3 in women). A high bleeding risk score (HAS-BLED ≥ 3) was noted in 17.4% of patients. Appendix A shows the clinical characteristics of patients according to stroke risk. 

### 3.2. Temporal Trends in Antithrombotic Therapy between 2004 and 2019 in the Total Study Population

Among the total study population, in most of the patients (81.1%), OAC therapy was used. Additionally, APT therapy was prescribed for 13.5% patients, heparins for 2.1% patients and 3.3% of patients did not receive any stroke prevention. Table 2 shows the stroke prevention strategy according to the stroke risk. 

OAC prescription significantly increased in the study period from 61.6% in 2004 to 97.4% in 2019 (Figure 2). Of those on OACs, most of the patients (59.2%) were treated with a VKA. Between 2004 and 2011, when NOACs were yet not approved, all patients who needed OACs were prescribed VKAs, but after the approval of NOACs, the use of VKAs decreased from 80.1% in 2012 to 20.7% in 2019. The percentage of patients treated with NOACs increased from 5.9% in 2012 to 79.3% in 2019 (Figure 3).

Among the patients on NOACs, 46.6% received dabigatran, 34.8% received rivaroxaban and 18.5% received apixaban. In the NOAC group, 39.4% of patients were treated with a reduced dose of NOACs. 

It was possible to observe a significant decrease in APT prescriptions from 28.6% in 2004 to 0.8% in 2019. At the same time, the proportion of patients not receiving any antithrombotic treatment decreased from 6.9% in 2004 to 0.4% in 2019. 

### 3.3. Temporal Trends in Antithrombotic Therapy between 2004 and 2019 in High Stroke Risk Patients 

Among high stroke risk patients, 81.6% were treated with OAC therapy, 13.5% with APT therapy, 2.1% with heparin and 2.8% of patients did not receive any antithrombotic therapy. 

The changes of particular antithrombotic regimens in the years 2004–2019 were as follows: OAC therapy—from 60.6% to 97.6%; APT therapy—from 29.1% to 0.6%; heparins—from 2.7% to 1.5% and no antithrombotic therapy—from 7.6% to 0.2% (Appendix A).

NOACs were prescribed for 33.2% of patients—4.8% in 2012 and 76.6% in 2019. Among patients on NOACs, 46% received dabigatran, 34.6% received rivaroxaban and 19.4% received apixaban. 

### 3.4. Temporal Trends in Antithrombotic Therapy between 2004 and 2019 in Intermediate Stroke Risk Patients

The antithrombotic therapy of intermediate stroke risk patients was similar to the therapy of high stroke risk patients. Most of the intermediate stroke risk patients (80.7%) were treated with OAC therapy, 13.8% with AP therapy, 1.4% with heparin and 4.2% of patients did not receive any antithrombotic therapy. 

A similar change of prescription of particular antithrombotic regimens in the years 2004–2019 were observed: OAC therapy—from 68.4% to 97.6%; APT therapy—from 24.6% to 0.2%; heparins—from 3.5% to 0% and no antithrombotic therapy—from 3.5% to 0.2% (Appendix A).

NOACs were prescribed for 33.2% of patients—11.9% in 2012 and 80.7% in 2019. Dabigatran was the most frequently chosen NOAC (51.8%), 34.7% of patients received rivaroxaban and 13.5% apixaban.

### 3.5. Temporal Trends in Antithrombotic Therapy between 2004 and 2019 in Low Stroke Risk Patients 

In patients without stroke risk factors, OAC therapy was prescribed to 70.9% of patients, APT therapy to 13% of patients, heparin to 3% of patients and 13% to patients did not receive any antithrombotic therapy. 

Temporal trends in stroke prevention were similar to these of the other stroke risk groups. Particular antithrombotic regimens in the years 2004–2019 were prescribed: OAC therapy—from 60.9% to 90.1%; APT therapy—from 30.4% to 0.3%; heparins—from 4.3% to 0.3% and no antithrombotic therapy—from 4.3% to 0.3% (Appendix A).

In patients with low stroke risk, 28.8% were treated with NOACs. Dabigatran, rivaroxban and apixaban were prescribed at the following frequencies: 48.1%, 39.6% and 12.3%. 

### 3.6. Factors Associated with OAC Use in the Total Study Population 

On the multivariate logistic regression analysis, the factors associated with OAC prescription were the period of hospitalization, previous thromboembolism, non-paroxysmal AF, hypertension, diabetes mellitus and hospitalization due to electrical cardioversion, and ablation or hospitalization due to AF without any procedure. Contrary to that, vascular disease, history of bleeding, cancer, anemia, thrombocytopenia and hospitalization due to acute coronary syndrome/percutaneous coronary intervention were associated with OAC non-prescription (Figure 4). 

## 4. Discussion

The present study provides an important view of contemporary antithrombotic therapy in AF patients, during the sixteen-year period when the AF guidelines concerning anticoagulant treatment were changed and a new antithrombotic therapy was introduced. The main findings of the present study are as follows: firstly, the prescription of OACs in stroke prevention significantly increased, and the percentage of patients treated with APT decreased; secondly, a similar percentage of patients with high, intermediate and low stroke risks were treated with OACs; and, thirdly, the factors predisposing the choice of OACs were identified, and part of them were not included in the CHA_2_DS_2_-VASc score. 

The prescription of OACs widely varies, depending on countries, study period and study populations. The Global Anticoagulant Registry in the Field with Atrial Fibrillation (GARFIELD-AF) reported temporal changes of antithrombotic therapy prescription patterns in AF patients, based on the comparison of cohorts between 2010 and 2015. In the present study, OAC prescriptions significantly increased in the study period, from 62% in 2004 to 97% in 2019. OAC prescription rates in the GARFIELD-AF study increased from 57% to 71% [11]. Similarly, the data from the Global Registry on Long-Term Oral Antithrombotic Treatment in Patients with Atrial Fibrillation (GLORIA-AF) also showed an increased OAC use [12]. In the GLORIA-AF study, the proportion of patients treated with OACs markedly increased from 64% to 80%, with the NOAC proportion greater than VKA. Lee SR et al. [13] showed that OAC prescriptions increased from 32% in 2008 to 46% in 2015. In the Australian population, the proportion of patients who were treated with OACs increased from 45% in 2009 to 72% in 2019 [14]. In our study, the percentage of patients treated with OACs in 2019 was extremely high, but a part of patients with a low thromboembolic risk were after electrical cardioversion or ablation due to AF and had a periodic indication to OAC treatment. 

A significant increase in OAC use in the recent years results from the introduction of NOACs. In the present study, the percentage of patients treated with NOACs increased from 5.9% in 2012 to 79.3% in 2019. This trend has also been observed in other studies. The NOAC use increased from 11.9% in 2011 to 94.0%, for all OAC initiations in 2019 [15]. 

The decrease in APT use is the main change in the current antithrombotic practice of stroke prevention in AF patients. In our study, during the sixteen-year period, it was possible to observe a significant decrease in APT prescriptions from 28.6% to 0.8%. At the same time, the proportion of patients not receiving any antithrombotic treatment decreased from 6.9% to 0.4%. It seems that, in the group of patients with the high risk of hemorrhagic complications, APT was applied for fear of potential hemorrhagic complications due to VKA usage. Therefore, NOACs, with a better safety profile than VKAs, became the pharmaceuticals of choice for patients who did not receive antithrombotic treatment earlier or received APT [7,8,9]. 

OAC use in the thromboembolic event prevention in AF patients should be conditioned by the thromboembolic risk [5]. In the present study, the percentage of OAC-treated patients in groups of high, intermediate and low thromboembolic risks was 81.6%, 80.7% and 70.9%, respectively. Interestingly, there was a high percentage of OAC-treated patients with low thromboembolic risk. In some of these patients, there were temporary indications to apply such OACs as electrical cardioversion or ablation. In the analysis of GARFIELD-AF, it was found that almost half of the patients with a CHA_2_DS_2_-VASc score equal to 0 (men) or 1 (women) received OACs [16]. In the Balkan Registry with 2712 patients included between 2014 and 2015, 56.5% truly low risk patients were recommended OACs [17]. In the PINNACLE Registry, 31.3% of patients without risk factors in the CHA_2_DS_2_-VASc score received OACs [18]. The GRASP-AF Registry showed that, from 2009 to 2018, the percentage of patients with low thromboembolic risk receiving OACs was 36.2–46.4% [19]. To explain the frequent use of OACs in patients with an AF of low thromboembolic risk, one might consider the significant limitations of the CHA_2_DS_2_-VASc score to assess thromboembolic risk. Although it was specifically constructed and validated for this purpose, it only captures a part of this risk. Therefore, the study population of patients with a CHA_2_DS_2_-VASc score equal to 0 (men) or 1 (women) was potentially augmented by emerging risk factors, such as chronic kidney disease, being overweight and a form of AF. This might explain why so many patients with a low thromboembolic risk, according to their CHA_2_DS_2_-VASc score, received OACs. 

In the present study, we showed that it is not only the factors included in the CHA_2_DS_2_-VASc score that influence the prescription of OACs. The strongest predictor of OAC use was the period of hospitalization; hospitalization between 2017 and 2018 was connected with a nearly 14-fold higher chance of receiving OACs than hospitalization between 2004 and 2006. This was due to the introduction of NOACs, safer than VKAs, to thromboembolic complication prophylaxis and contraindication for APT use in thromboembolic complication prophylaxis in AF patients. A significant predictor of using OACs was also the type of AF. Although the available data have not consistently demonstrated the association of the type of AF with the risk of thromboembolic complications, several studies have shown a higher risk of thromboembolism in patients with permanent AF than persistent or paroxysmal AF [20]. In addition, Ganesan et al. showed that non-paroxysmal AF increases the risk of thromboembolic complications by 38%, compared to paroxysmal AF [21]. In our study, non-paroxysmal AF (persistent or permanent) increased the chance of OAC prescription almost three-fold. The factors diminishing the chance to prescribe OACs were vascular disease; factors increasing the hemorrhage risk, i.e., hemorrhage in the medical history; neoplastic disease; thrombocytopenia; anemia and acute coronary syndrome/PCI as a reason for being admitted to hospital. Similarly, Lee et al. [13] showed that the presence of vascular disease and prior intracranial hemorrhage were associated with OAC underuse. To date, the binding guidelines point to the idea that the hemorrhage risk should not be the reason for using OACs. What is more, the high risk of hemorrhagic complications significantly limits OAC applications in clinical practice.

## 5. Study Strengths and Limitations 

The present study includes unique descriptions of clinical data from the Polish AF populations, rather than data from selected or registered patients from trials. Our findings reflect the real-world clinical practice pattern of antithrombotic strategies in AF patients. Several limitations related to the retrospective nature of the data used should be underlined. First of all, due to the lack of long-term observation of the patients, it is not possible to evaluate a long-term prognosis for patients with AF treated with an individual antithrombotic strategy. Secondly, in the present study, hospitalized patients with AF were assessed; among them, only some had a first-time diagnosed AF and only for them an anticoagulant therapy was initiated. Thus, despite the registry referring to hospitalized patients, the anticoagulant therapy used for most of them was initiated in ambulatory conditions before admitting them to hospital.

## 6. Conclusions

During the sixteen-year study period, a significant increase in OAC use and a decrease in AP use was noted. After the approval of NOACs, the use of VKAs significantly decreased. In patients of high and intermediate stroke risks, the prevention of stroke was very similar; for patients with a low stroke risk, OAC prescription was lower than for others. Factors other than these included in the CHA_2_DS_2_-VASc score were independent predictors of OAC use.

## Figures and Tables

**Figure 1 ijerph-19-05584-f001:**
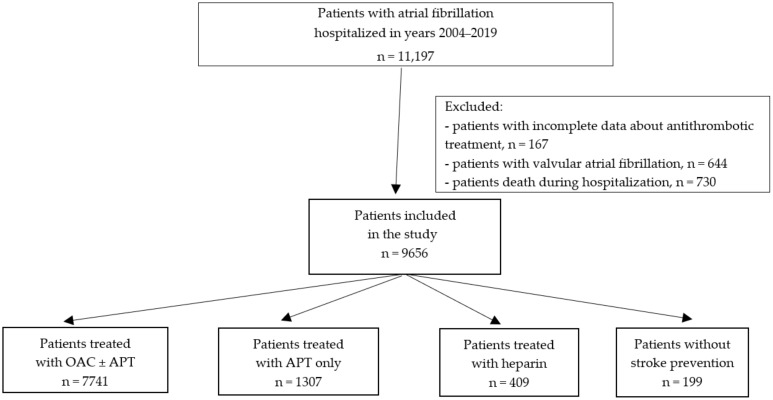
The flow chart of the study. Abbreviations: APT, antiplatelet drug, and OAC, oral anticoagulant.

**Figure 2 ijerph-19-05584-f002:**
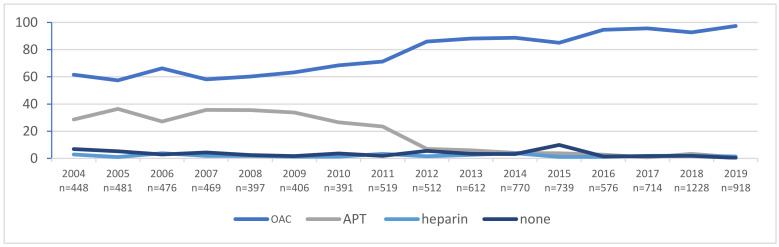
Temporal trends of antithrombotic therapy in all study patients. Abbreviations: APT, antiplatelet drug, and OAC, oral anticoagulant.

**Figure 3 ijerph-19-05584-f003:**
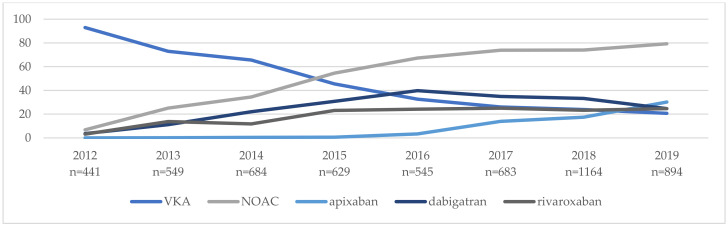
Temporal trends of anticoagulant therapy in all study patients in the years 2012–2019 (after NOACs approval). Abbreviations: NOAC, non-vitamin K oral anticoagulant, and VKA, vitamin K antagonist.

**Figure 4 ijerph-19-05584-f004:**
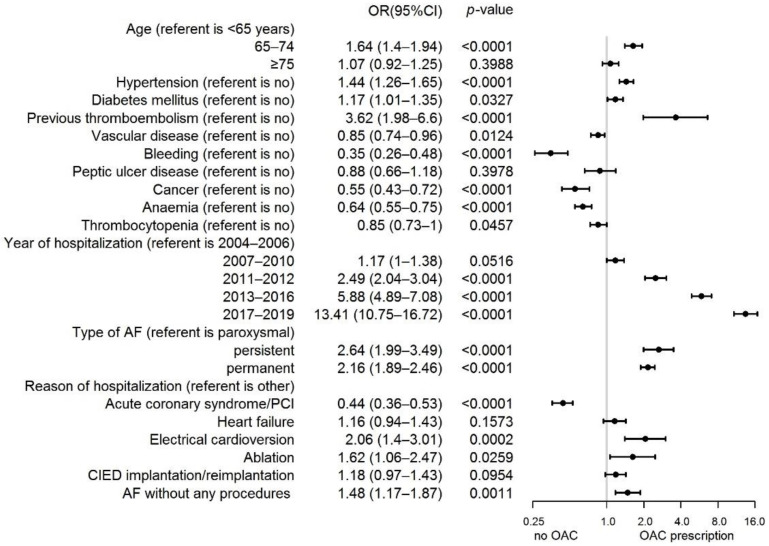
Factors associated with the oral anticoagulants prescriptions for stroke prevention in patients with AF: multivariable logistic regression models. Abbreviations: AF, atrial fibrillation; CIED, cardiac implantable electronic device; OAC, oral anticoagulant; OR, odds ratio; and PCI, percutaneous coronary intervention.

**Table 1 ijerph-19-05584-t001:** Baseline characteristic of the study group according to stroke prevention.

Clinical Characteristic	All n = 9656	OAC Therapy n = 7827	No OAC Therapy n = 1829	APT Therapy n = 1307	Heparin Therapy n = 199	No Therapy n = 323
Type of Atrial Fibrillation						
Paroxysmal	4389 (45.5)	3269 (41.8)	1120 (61.2)	825 (63.1)	95 (47.7)	200 (61.9)
Persistent	1439 (14.9)	1343 (17.2)	96 (5.3)	59 (4.5)	13 (6.5)	24 (7.4)
Permanent	3828 (39.6)	3215 (41.1)	613 (33.5)	423 (32.4)	91 (45.7)	99 (30.7)
**Stroke risk factors**						
Age, years						
Mean (SD)	71.2 (11.2)	71.2 (10.9)	71.3 (12.4)	71.8 (11.4)	72.3 (11.6)	68.4 (16.0)
Median (IQR)	72 (64–80)	72 (64–79)	73 (63–81)	74 (64–80)	75 (64.5–80.5)	71 (60–81)
<65	2501 (25.9)	1984 (25.3)	517 (28.3)	349 (26.7)	50 (25.1)	118 (36.5)
65–74	3055 (31.6)	2593 (33.1)	462 (25.3)	343 (26.2)	45 (22.6)	74 (22.9)
≥75	4100 (42.5)	3250 (41.5)	850 (46.5)	615 (47.1)	104 (52.3)	131 (40.6)
Female	4221 (43.7)	3419 (43.7)	802 (43.9)	571 (43.7)	86 (43.2)	145 (44.9)
Heart failure	5667 (58.7)	4613 (58.9)	1054 (57.6)	731 (55.9)	135 (67.8)	188 (58.2)
Hypertension	7387 (76.5)	6085 (77.7)	1302 (71.2)	962 (73.6)	139 (69.8)	201 (62.2)
Previous stroke/ TIA/peripheral embolism	1266 (13.1)	1072 (13.7)	194 (10.6)	134 (10.3)	36 (18.1)	24 (7.4)
Diabetes mellitus	2463 (25.5)	2061 (26.3)	402 (22.0)	293 (22.4)	49 (24.6)	60 (18.6)
Vascular disease	3364 (34.8)	2660 (34)	704 (38.5)	555 (42.5)	73 (36.7)	76 (23.5)
**Medical history**						
Bleeding	267 (1.9)	188 (2.4)	79 (4.3)	42 (3.2)	14 (7)	23 (7.1)
Cancer	408 (4.2)	293 (3.7)	115 (6.3)	60 (4.6)	36 (18.1)	19 (5.9)
Peptic ulcer disease	326 (3.4)	243 (3.1)	83 (4.5)	57 (4.4)	6 (3)	20 (6.2)
Anemia	1629 (16.9)	1267 (16.2)	362 (19.8)	214 (16.4)	64 (32.2)	84 (26)
Thrombocytopenia	1457 (15.1)	1158 (14.8)	299 (16.3)	196 (15)	42 (21.1)	61 (18.9)
eGFR < 60 mL/min/1.73 m^2^	6043 (62.6)	4925 (62.9)	1118 (61.1)	807 (61.7)	120 (60.3)	191 (59.1)
eGFR, mean (SD)	55.2	55.1	55.3	55.8	54.4	55.1
NSAID use	73 (0.8)	45 (0.6)	14 (0.8)	11 (0.8)	1 (0.5)	2 (0.6)
Abnormal liver function	112 (1.2)	66 (0.8)	23 (1.3)	8 (0.6)	7 (3.5)	8 (2.5)
Alcohol abuse	178 (1.8)	104 (1.3)	37 (2.0)	26 (1.9)	2 (1.0)	9 (2.8)
**Thromboembolism risk**						
CHADS_2_						
Mean (SD)	2.3 (1.3)	2.3 (1.3)	2.2 (1.3)	2.2 (1.3)	2.5 (1.5)	1.9 (1.4)
Median (IQR)	2 (1–3)	2 (1–3)	2 (1–3)	2 (1–3)	2 (1.5–3)	2 (1–3)
CHA_2_DS_2_VASc						
Mean (SD)	3.8 (1.8)	3.8 (1.8)	3.7 (1.9)	3.8 (1.8)	4.1 (1.9)	3.3 (2.0)
Median (IQR)	4 (3–5)	4 (3–5)	4 (2–5)	4 (2–5)	4 (3–5)	4 (2–5)
CHA_2_DS_2_VASc = 0 in men, 1 in women	368 (3.8)	261 (3.3)	107 (5.9)	48 (3.7)	11 (5.5)	48 (14.9)
CHA_2_DS_2_VASc = 1 in men, 2 in women	1024 (10.6)	826 (10.6)	198 (10.8)	141 (10.8)	14 (7)	43 (13.3)
CHA_2_DS_2_VASc ≥ 2 in men, 3 in women	8264 (85.6)	6740 (86.1)	1524 (83.3)	1118 (85.5)	174 (87.4)	232 (71.8)
**Bleeding risk**						
HAS-BLED						
Mean (SD)	1.8 (0.9)	1.8 (0.9)	1.6 (0.9)	1.6 (0.9)	2 (1.0)	1.6 (1.1)
Median (IQR)	2 (1–2)	2 (1–2)	2 (1–2)	2 (1–2)	2 (1–2)	2 (1–2)
HAS-BLED ≥ 3	1676 (17.4)	1451 (18.5))	227 (12.4)	121 (9.3)	47 (23.6)	59 (18.3)
**Reason for hospitalization**						
CIED implantation/ reimplantation	2207 (22.9)	1708 (21.8)	499 (27.3)	378 (28.9)	47 (23.6)	74 (22.9)
Heart failure	1998 (20.7)	1707 (21.8)	291 (15.9)	174 (13.3)	49 (24.6)	68 (21.1)
Acute coronary syndrome/planned PCI	1213 (12.6)	700 (8.9)	513 (28.1)	468 (35.8)	17 (8.5)	28 (8.7)
AF without any procedures	1122 (11.6)	940 (12)	182 (9.9)	97 (7.4)	21 (10.6)	64 (19.8)
Electrical cardioversion	1056 (10.9)	997 (12.7)	59 (3.2)	31 (2.4)	11 (5.5)	17 (5.3)
Ablation	339 (3.5)	308 (3.9)	31 (1.7)	17 (1.3)	2 (1.0)	12 (3.7)
Other	1721 (17.8)	1467 (18.7)	254 (13.9)	142 (10.9)	52 (26.1)	60 (18.6)
**Years of hospitalization**						
2004–2006	1405 (14.6)	867 (11.1)	538 (29.4)	432 (33.1)	36 (18.1)	70 (21.7)
2007–2010	1663 (17.2)	1037 (13.2)	626 (34.2)	549 (42)	26 (13.1)	51 (15.8)
2011–2012	1031 (10.7)	810 (10.3)	221 (12.1)	158 (12.1)	25 (12.6)	38 (11.8)
2013–2016	2697 (27.9)	2397 (30.6)	300 (16.4)	113 (8.6)	61 (30.7)	126 (39.0)
2017–2019	2860 (29.6)	2716 (34.7)	144 (7.9)	55 (4.2)	51 (25.6)	38 (11.8)

The numbers are presented as the mean (standard deviation), median (interquartile range) or numbers (percentage) otherwise mentioned. Abbreviations: AF, atrial fibrillation; APT, antiplatelet drug; CIED, cardiac implantable electronic device; eGFR, estimated glomerular filtration rate; NSAID, *Non-Steroidal Anti-Inflammatory Drug;* IQR, interquartile range; PCI, percutaneous coronary intervention; SD, standard deviation; and TIA, transient ischemic attack. CHA_2_DS_2_-VASc score: congestive heart failure (1 point), hypertension (1 point), age: ≥75 years (2 points), diabetes mellitus (1 point), stroke/TIA/thromboembolism (2 points), vascular disease (1 point), age: 65–74 years (1 point) and sex: female (1 point). HAS-BLED score: hypertension (1 point), liver disease (1 point), renal disease (1 point), stroke history (1 point), bleeding history (1 point), age: >65 years (1 point) and drug (concomitant use of NSAID or antiplatelet agent, 1 point).

**Table 2 ijerph-19-05584-t002:** Stroke prevention according to stoke risk in the study group.

Stroke Prevention	All n = 9656	High Stroke Risk n = 8264	Intermediate Stroke Risk n = 1024	Low Stroke Risk n = 368	*p*
OAC	7827 (81.1)	6740 (81.6)	826 (80.7)	261 (70.9)	<0.0001
VKA	4637 (48)	3996 (48.4)	486 (47.5)	48 (13)	<0.0001
NOAC	3190 (33)	2744 (33.2)	340 (33.2)	155 (42.1)	0.0018
APT	1307 (13.5)	1118 (13.5)	141 (13.8)	106 (28.8)	<0.0001
Heparin	199 (2.1)	174 (2.1)	14 (1.4)	11 (3)	0.1291
None	323 (3.3)	232 (2.8)	43 (4.2)	48 (13)	<0.0001

Abbreviations: APT, antiplatelet drug; NOAC, non-vitamin K oral anticoagulant; OAC, oral anticoagulant; and VKA, vitamin K antagonist.

## Data Availability

Data is available on request for the first author.

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
