# Peer review of "Temporal Trends in Oral Anticoagulant Prescription in Atrial Fibrillation Patients between 2004 and 2019"

_ijerph, 2022, doi:10.3390/ijerph19095584_

Round 1
Reviewer 1 Report
The study is interesting and is based on an assessment of treatment trends
over a 15-year period, in a large patient population. It has the typical
disadvantages of a retrospective study, so we do not know what the
effectiveness of a given therapy is, whether it has not been changed after
some time, whether the patient was taking prescribed medications.
In my opinion, the article requires a more precise definition of some things:
1. Does not issuing a prescription equal no treatment recommendation?
2. What influenced the type of prescribed therapy, the article describes what
should influence the decision whether a patient will be prescribed anticoagulant therapy, but it is not clear what influenced the type of therapy prescribed?
Author Response
Dear Reviewer,
I am pleased to resubmit for publication the revised version of Temporal Trends in Oral Anticoagulant Prescription in Atrial Fibrillation Patients between 2004 and 2019.
The Reviewers’ comments were very helpful and greatly appreciated. We have addressed each concern and hope that this revised manuscript is now acceptable. Each comment is discussed in detail below. We made corrections according to reviewers’ suggestions and all the corrections were made using the Track Changes function. Thank you for allowing us to resubmit our manuscript for your consideration.
The specific responses to the Reviewers’ comments are as follows:
Reviewer 1
In my opinion, the article requires a more precise definition of some things:
- Does not issuing a prescription equal no treatment recommendation?
The presented study evaluated the anticoagulant treatment recommended by physicians in patients with atrial fibrillation. The degree of compliance with medical recommendations was not assessed. Thank you for your remark, we will plan a study in which we will assess the degree of implementation of the anticoagulant therapy recommendations.
- What influenced the type of prescribed therapy, the article describes what
should influence the decision whether a patient will be prescribed anticoagulant therapy, but it is not clear what influenced the type of therapy prescribed?
In the presented study, the factors predisposing patients to anticoagulant treatment prescription were assessed. It was shown that factors associated with oral anticoagulant prescription were period of hospitalization, previous thromboembolism, non-paroxysmal atrial fibrillation, hypertension, diabetes mellitus and hospitalization due to electrical cardioversion, ablation or hospitalization due to atrial fibrillation without any procedure. Contrary to that, vascular disease, history of bleeding, cancer, anaemia, thrombocytopenia and hospitalization due to acute coronary syndrome / percutaneous coronary intervention were associated with oral anticoagulant non-prescription.
We hope the above-listed revisions and answers are satisfactory and will meet your expectations.
Once again thank you for reviewing our manuscript.
Yours faithfully,
Authors
Reviewer 2 Report
The authors submit a retrospective, observational single-center analysis on almost 10.000 patients admitted for atrial fibrillation. Focus is on (1) trends of antithrombotic therapy over a time interval of 15 years and (2) to detect possible parameters associated with use of oral anticogulants (OAC). Main findings are a significant increase in the rate of OAC and detection of non-CHADS-VASC score parameters associated with use of oral anticoagulants.
The paper is well structured, English language and spelling are fine.
Major concern arises from the observational nature of the work: factors associated with OAC use should be viewed as coincidental by principle. No causal relations can be proven. Thus, wording should be very cautious when discussing these factors. Furthermore, no information on concomitant antiarrhythmic therapy is provided and no information on the basic therapeutic principles of rhythm vs. rate control. If the latter could be supplemented, this would greatly enhance the overall merit of the work.
At least however, these possible confounders should be clearly discussed.
Author Response
Dear Reviewer,
I am pleased to resubmit for publication the revised version of Temporal Trends in Oral Anticoagulant Prescription in Atrial Fibrillation Patients between 2004 and 2019.
The Reviewers’ comments were very helpful and greatly appreciated. We have addressed each concern and hope that this revised manuscript is now acceptable. Each comment is discussed in detail below. We made corrections according to reviewers’ suggestions and all the corrections were made using the Track Changes function. Thank you for allowing us to resubmit our manuscript for your consideration.
The specific responses to the Reviewers’ comments are as follows:
Major concern arises from the observational nature of the work: factors associated with OAC use should be viewed as coincidental by principle. No causal relations can be proven. Thus, wording should be very cautious when discussing these factors.
We absolutely agree with the comment. An observational, retrospective study was conducted to identify factors that influence the recommended anticoagulant treatment. While discussing the identified factors, we were aware that the choice of the recommended anticoagulant therapy could also be influenced by geographic factors or study period.
Furthermore, no information on concomitant antiarrhythmic therapy is provided and no information on the basic therapeutic principles of rhythm vs. rate control. If the latter could be supplemented, this would greatly enhance the overall merit of the work.
The presented study was designed to assess trends in anticoagulant treatment. We do not have data on antiarrhythmic therapy in the study group because it was not the subject of this study. We are preparing a study using the data from the POL-AF registry, a prospective, multicenter registry of patients with atrial fibrillation, where we will present issues related to antiarrhythmic therapy.
We hope the above-listed revisions and answers are satisfactory and will meet your expectations.
Once again thank you for reviewing our manuscript.
Yours faithfully,
Authors
Reviewer 3 Report
The authors present an interesting retrospective study aimed at reviewing the prescription of anticoagulants in atrial fibrillation patients on a massive caseload and a long timespan at a single center facility.
The description of the method is clear and concise, with sufficient details to understand the methods and the rationale.
The language is correct, the paper is well written, every step needed to present a formally correct work are present, the statistical implant is acceptable and kept to a minimum, as appropriate for this kind of study. The study population appears robust enough to support the conclusions. The graphs are beautiful and very descriptive.
The conclusions are straightforward and supported by evidence with a reasonably large sample to support the observations.
The interest of the paper is somehow significant to the surgeons audience, but the paper suffers from a lack of readability mainly due to the imbalance between text and images.
Please consider adding a central message graph to synthetize the message and consider appropriate graphs to convey part of the message In the tables, most reader would greatly appreciate it.
Author Response
Dear Reviewer,
I am pleased to resubmit for publication the revised version of Temporal Trends in Oral Anticoagulant Prescription in Atrial Fibrillation Patients between 2004 and 2019.
The Reviewers’ comments were very helpful and greatly appreciated. We have addressed each concern and hope that this revised manuscript is now acceptable. Each comment is discussed in detail below. We made corrections according to reviewers’ suggestions and all the corrections were made using the Track Changes function. Thank you for allowing us to resubmit our manuscript for your consideration.
The specific responses to the Reviewers’ comments are as follows:
The authors present an interesting retrospective study aimed at reviewing the prescription of anticoagulants in atrial fibrillation patients on a massive caseload and a long timespan at a single center facility.
The description of the method is clear and concise, with sufficient details to understand the methods and the rationale.
The language is correct, the paper is well written, every step needed to present a formally correct work are present, the statistical implant is acceptable and kept to a minimum, as appropriate for this kind of study. The study population appears robust enough to support the conclusions. The graphs are beautiful and very descriptive.
The conclusions are straightforward and supported by evidence with a reasonably large sample to support the observations.
The interest of the paper is somehow significant to the surgeons audience, but the paper suffers from a lack of readability mainly due to the imbalance between text and images.
Please consider adding a central message graph to synthetize the message and consider appropriate graphs to convey part of the message In the tables, most reader would greatly appreciate it.
Thank you for your feedback on our manuscript. The tables have been supplemented with clinically relevant information. As a central illustration, we consider fig. 4 showing the factors associated with the oral anticoagulant prescriptions for stroke prevention in patients with atrial fibrillation: multivariable logistic regression models.
Numerous graphical data have been added in the supplement materials.
We hope the above-listed revisions and answers are satisfactory and will meet your expectations.
Once again thank you for reviewing our manuscript.
Yours faithfully,
Authors
Reviewer 4 Report
The authors aimed to assess temporal trends of antithrombotic therapy and identify factors predisposing to oral anticoagulant (OAC) use in stroke
prevention in AF patients. They concluded that in hospitalized patients with AF, during sixteen years of the study period, a significant increase of OAC use and a decrease of APT use were noted. Factors other
than these included in CHA2DS2-VASc score were independent predictors of OAC use.
I have the following concerns:
- Please mention the 'A' component of the ABC pathway in context of indications to OAC. Please define stroke risk according to CHA2DS2-VASc score in patients who have indications to OAC.
- The specific terms used in tables, for example bleeding, should be defined.
- Please define high, intermediate, and low stroke risk.
- Please include NSAIDs use, alcohol abuse, abnormal liver function, median creatinine clearance in Table 1.
- What are the practical implications of the study?
- Could you please include NOAC and VKA in trends of antithrombotic therapy in conclusions?
- What were the reasons for not using OAC in patients?
- What was the proportion of patients with CHA2DS2-VASc score more than 2 taking OAC in time?
Author Response
Dear Reviewer,
I am pleased to resubmit for publication the revised version of Temporal Trends in Oral Anticoagulant Prescription in Atrial Fibrillation Patients between 2004 and 2019.
The Reviewers’ comments were very helpful and greatly appreciated. We have addressed each concern and hope that this revised manuscript is now acceptable. Each comment is discussed in detail below. We made corrections according to reviewers’ suggestions and all the corrections were made using the Track Changes function. Thank you for allowing us to resubmit our manuscript for your consideration.
The specific responses to the Reviewers’ comments are as follows:
1. Please mention the 'A' component of the ABC pathway in context of indications to OAC. Please define stroke risk according to CHA2DS2-VASc score in patients who have indications to OAC.
The data concerning ABC pathway and stroke risk according to the CHA2DS2-VASc score were added.
2. The specific terms used in tables, for example bleeding, should be defined.
In the Supplementary Materials, the data on terms used in the tables were added (Table S1).
3. Please define high, intermediate, and low stroke risk.
In the Methodology section, high, intermediate, and low stroke risk definitions were added.
4. Please include NSAIDs use, alcohol abuse, abnormal liver function, median creatinine clearance in Table 1.
According to the Reviewer’s suggestion, we included the data on NSAID use, alcohol abuse, abnormal liver function, median creatine clearance in Table 1.
5. What are the practical implications of the study?
Thank you for this remark. The present study provides an important view of contemporary antithrombotic therapy in AF patients during the sixteen-year period when the AF guidelines concerning anticoagulant treatment were changed and a new antithrombotic therapy was introduced. The main findings of the present study are as follows. Firstly, prescription of OACs in stroke prevention increased significantly, and the percentage of patients treated with APT decreased. Secondly, similar percentage of patients with high, intermediate and low stroke risk were treated with OACs. Thirdly, factors predisposing the choice of OACs were identified and a part of them was not included in CHA2DS2-VASc score. In our opinion the last conclusion has the most considerable clinical and practical implication. According to the current guidelines, only stroke risk (assessment according to CHA2DS2-VASc score) should decide whether to apply OACs. The data from numerous observational studies, including the present study, show that factors other than the CHA2DS2-VASc score determine the initiation of anticoagulant treatment. It seems that practitioners take into account also other factors increasing the risk of thromboembolic complications, and not included in the CHA2DS2-VASc score, i.e. form of AF, chronic kidney disease.
6. Could you please include NOAC and VKA in trends of antithrombotic therapy in conclusions?
According to the Reviewer’s suggestion, we included NOAC and VKA trends in Conclusions.
7. What were the reasons for not using OAC in patients?
Thank you for this question. In the multivariate logistic regression analysis, we showed that vascular disease, history of bleeding, cancer, anaemia, thrombocytpenia and hospitalization due to acute coronary syndrome/ percutaneous coronary intervention were associated with OAC non-prescription.
8. What was the proportion of patients with CHA2DS2-VASc score more than 2 taking OAC in time?
In the Results section, the data on percetnage of patients with CHA2DS2-VASc score more than 2 treated with OACs were added (Section 3.3. and figure S1.).
Percentage o patients treated with OAC in each year was as follow:
2004-60.6%
2005-58.5%
2006-66.4%
2007-57.8%
2008-59.3%
2009-63.5%
2010-67.4%
2011-71.1%
2012-85.5%
2013-89.2%
2014-88.8%
2015-85.8%
2016-95%
2017-97.1%
2018-93.4%
2019-97.6%
We hope the above-listed revisions and answers are satisfactory and will meet your expectations.
Once again thank you for reviewing our manuscript.
Yours faithfully,
Authors
Round 2
Reviewer 2 Report
The authors have adequately responded to comments and improved the manuscript accordingly.
Reviewer 4 Report
Thank your reply. I have no further comments